# Local Beneficial Microorganisms Impact Carbon and Nitrogen Mineralization in a Lixisol Incubated with Organic Waste Products

Emmanuel Noumsi-Foamouhoue [1,2,3,*], Samuel Legros [1,2,4], Paula Fernandes [2,5,6], Laurent Thuriès [1,4], Komi Assigbetsé [2,7], Aboubacry Kane [3,8], Frédéric Feder [1,4] and Jean-Michel Médoc [1,2,4,*]

1   UPR Recyclage et Risque, CIRAD, F-34398 Montpellier, France; samuel.legros@cirad.fr (S.L.); frederic.feder@cirad.fr (F.F.)
2   Laboratoire Mixte International IESOL, ISRA-IRD Bel-Air Center, Dakar BP 1386, Senegal; paula.fernandes@cirad.fr (P.F.); komi.assigbetse@ird.fr (K.A.)
3   FST, Département de Biologie Végétale, UCAD, Dakar BP 1386, Senegal; aboubacry.kane@ucad.edu.sn
4   Recyclage et Risque, University Montpellier, CIRAD, F-34090 Montpellier, France
5   UPR HortSys, CIRAD, F-34398 Montpellier, France
6   HortSys, University Montpellier, CIRAD, F-34090 Montpellier, France
7   Institut de Recherche pour le Développement (IRD), F-34394 Montpellier, France
8   Laboratoire Commun de Microbiologie (LCM), IRD-ISRA-UCAD Bel-Air Center, Dakar BP 1386, Senegal
*   Correspondence: emmanuelnofoma@yahoo.com or emmanuel.noumsi_foamouhoue@cirad.fr (E.N.-F.); jean-michel.medoc@cirad.fr (J.-M.M.)

**Abstract:** Growing awareness of the environmental impact of intensive agriculture has prompted a quest for more sustainable approaches. The most promising alternatives include the application of organic waste products (OWPs), as well as biofertilizers containing local beneficial microorganisms (BMs) on cultivated soils. This study was designed to assess the effects of BMs on carbon and nitrogen mineralization of OWPs. A 28-day laboratory incubation experiment was conducted at 28 °C with a soil, three OWPs (poultry litter (PL), cow dung (CD), and sewage sludge (SS)), and three BMs (groundnut + millet from Saint-Louis (LGM), groundnut from the southern groundnut basin (BG), and rice from the southern groundnut basin (BR) in Senegal), alone and combined. The results showed that the C mineralization from OWP + BM + soil mixtures exceeded (range 13–41%) those measured for OWP + soil. The BM input induced an increase or reduction in OWP nitrogen mineralization, depending on the type of BM and OWP. However, the net mineral nitrogen (Nmin) obtained with the PL-LGM and SS-BG combinations was 13.6- and 1.7-fold higher than with PL and SS, respectively, at 28 days. The addition of BM seemed to lead to a decrease in the C: N ratio, an improvement in the availability of nitrogen, and an increase in microbial activity in the OWP + BM + soil mixture. Our results generated new information on the variation patterns of OWP carbon and nitrogen in OWP-BM-soil systems. This novel insight will be developed to guide the most appropriate choice of OWP-BM mixtures for improved fertilization in sustainable production systems.

**Keywords:** carbon mineralization dynamics; nitrogen mineralization dynamics; organic waste; biofertilizers; indigenous beneficial microorganisms; soil fertility; plant nutrition; sustainable agriculture

## 1. Introduction

Intensive agriculture is quantitatively efficient but has its drawbacks in terms of environmental impact. The decline in organic matter content in intensively cropped soils and the heavy use of mineral fertilizers in intensive cropping systems are major shortcomings [1–3]. This, in turn, may lead to (i) agricultural soil degradation (e.g., hardening and low nutrient content and biodiversity); (ii) pollution of groundwater, which is the primary source of drinking water; and (iii) substantial emissions of nitrous oxide ($N_2O$), a potent

greenhouse gas [4–7]. It is therefore crucial to design sustainable agricultural production systems to maintain soil fertility while safeguarding the environment.

One known solution to this issue involves the recycling of organic waste products, whose production is constantly increasing [8–11]. OWP encompasses all organic waste and by-products from anthropogenic and agricultural activities [12]. These residues can be of animal, plant, urban, industrial, or agro-industrial origin [13,14]. The impacts of OWP inputs on soil physicochemical and biological properties have been documented. The results have revealed that OWP application can have beneficial effects, such as improving soil porosity, while increasing soil microflora and fauna activity [15–17]. Moreover, OWP use can modify nutrient cycling via carbon and nitrogen mineralization [18,19].

Indeed, previous studies have shown that the application of OWP, including livestock manure, sewage sludge, livestock waste, and crop residues, on various soil types (soil + OWP mixtures) can result in an increase in soil C-$CO_2$ and an increase/decrease in soil mineral N [20–23]. For instance, the results of Santos et al. [24] showed high carbon mineralization with significantly higher $CO_2$ emissions in clay soils amended with different types of composted agro-industrial waste, compared to an unamended soil, throughout a 55-day incubation at 20 °C. The results of Pansu and Thuriès [20] showed a significant increase/decrease in mineral N in a sandy soil amended with animal waste, manure, or compost, compared with the unamended soil, during a 180-day incubation at 28 °C. It has been shown that these different carbon and nitrogen mineralization phenomena depend on several factors, including (i) the physicochemical properties of the soil and OWP (e.g., water content, organic carbon (OC) and organic nitrogen (ON) content, carbon-to-nitrogen (C: N) ratio, dissolved organic carbon (DOC) content, and pH) [25,26], (ii) the biological properties of the soil and OWP (e.g., microbial diversity and abundance) [27,28], and (iii) the OWP decomposability [29,30]. For instance, organic waste with a low C: N (<20) shows greater N mineralization compared to the residues with wide C: N ratios, which cause N immobilization [31]. Khalil et al. [32] stated that the significant increase in ammonification observed with the application of chicken manure could be attributed to the increase in pH, higher amounts of N, a low C: N ratio, and low lignin content. Khalil et al. [33] have shown that the addition of organic waste can raise soil pH, thus promoting carbon mineralization in acid soils. The high concentration of readily biodegradable organic carbon (or DOC) introduced with the application of an organic product stimulates microcosm microbial activity [34,35].

Another recognized option for preserving soil and environmental health involves the application of biofertilizers containing local beneficial microorganisms in agricultural production systems [36–38]. BMs are local biological inocula derived from leaf litter collected on forest soils in the vicinity of the sites where they are to be applied [37]. This input contains a range of microbial communities, such as photosynthetic bacteria, lactic acid bacteria, actinomycetes, and fungi [38,39]. The impacts of BM inputs on soil quality and crop production have been documented in the literature. The findings highlighted beneficial effects related to BM application, such as (i) increased beneficial soil microorganism abundance and diversity [39]; (ii) improved soil structure [40]; (iii) increased soil nutrient content (e.g., potassium, calcium, and zinc) [41]; (iv) elimination of soilborne pathogens and diseases [42,43]; (v) enhanced plant growth and development [44,45]; and (vi) increased crop yield and quality [46,47]. In addition, certain microbial communities in BM can decompose soil organic matter [48,49].

Indeed, the results of some studies reported in the literature show the ability of certain microbial communities to mineralize soil carbon and/or nitrogen. These include Actinobacteria, which are capable of degrading polysaccharides, osamines, cellulose, lignocellulose, and lignin via a range of enzymes (β-glucosidase, xylanase, protease, cellobiohydrolase, cellulases, hemicelluloses, and other ligninolytic enzymes) [50,51]. Ling et al. [52] highlighted the key role of Bacteroidia and Proteobacteria in the mineralization of carbon bound to complex soil compounds. Firmicutes, Actinobacteria, and Ascomycota have been reported to be associated with organic nitrogen degradation and nitrogen cycling in previous

research [48,49,52]. Certain bacteria, such as *Bradyrhizobium* and *Bacillus*, have also been examined in some studies in the literature, illustrating their strong ability to reduce and dissolve nitrogen components through denitrification [53,54]. The decomposition of soil organic matter may also depend on the growth of fungi such as Ascomycota, which have a major role in carbon mineralization [55].

Although factors related to carbon and nitrogen dynamics in soil + OWP mixtures have been demonstrated, few studies show the effects of biofertilizer input containing local beneficial microorganisms on carbon and nitrogen dynamics in soil + OWP + BM mixtures. Our hypothesis postulates that the addition of BM leads to excess mineralization of carbon and nitrogen from OWP in soil + OWP + BM mixtures.

This study aimed to assess the effects of BM input on carbon and nitrogen mineralization kinetics of OWP in soil + OWP + BM mixtures. We thus measured C-CO$_2$ and Nmin in an incubation experiment under controlled conditions, using soil and different types of OWP and BM, both alone and combined.

## 2. Materials and Methods

### 2.1. Soil

The soil (S) used in this study is a tropical ferruginous leached sandy loam soil classified as a Lixisol [56,57]. It was sampled at the Institut Sénégalais de Recherches Agricoles (ISRA) research station (13°45′29″ N, 15°47′12″ W) located at Nioro du Rip in the southern groundnut basin in Senegal. Composite samples were obtained at a 0–30 cm depth on a plot that had been fallowed for 3 years. The soil samples were air-dried and sieved (at 2 mm) before the incubation experiment. The main soil physicochemical properties are presented in Table 1.

**Table 1.** Main soil, OWP, and BM properties.

| | Soil | CD | PL | SS | LGM | BG | BR |
|---|---|---|---|---|---|---|---|
| TOC (g 100 g$^{-1}$ DM) | 0.28 | 31.99 | 27.36 | 23.75 | 28.99 | 34.24 | 23.83 |
| TON (g kg$^{-1}$ DM) | 0.20 | 19.70 | 65.00 | 25.50 | 11.00 | 9.70 | 6.90 |
| N-NO$_3$ (mg kg$^{-1}$ DM) | 0.80 | n.d. | n.d. | n.d. | n.d. | n.d. | n.d. |
| N-NH$_4$ (mg kg$^{-1}$ DM) | 5.40 | n.d. | n.d. | n.d. | n.d. | n.d. | n.d. |
| C: N | 13.00 | 16.24 | 4.21 | 9.31 | 26.35 | 35.30 | 34.54 |
| Total P (mg kg$^{-1}$ DM) | 65 | 3493 | 12,941 | 9274 | 1164 | 580 | 452 |
| Assim. P (mg kg$^{-1}$ DM) | 7 | 762 | 1141 | 196 | 360 | 192 | 162 |
| CEC (cmol(+) kg$^{-1}$ DM) | 1.47 | n.d. | n.d. | n.d. | n.d. | n.d. | n.d. |
| WC (g 100 g$^{-1}$) | 3.32 | 68.74 | 8.12 | 3.74 | 56.66 | 59.00 | 64.12 |
| pH$_{(H_2O)}$ | 6.67 | 7.65 | 7.57 | 6.98 | 7.09 | 7.28 | 7.87 |
| Sand (g 100 g$^{-1}$ DM) | 83.50 | n.d. | n.d. | n.d. | n.d. | n.d. | n.d. |
| Silt (g 100 g$^{-1}$ DM) | 9.50 | n.d. | n.d. | n.d. | n.d. | n.d. | n.d. |
| Clay (g 100 g$^{-1}$ DM) | 5.40 | n.d. | n.d. | n.d. | n.d. | n.d. | n.d. |

Data presented for soil, OWP, and BM, dried at 105 °C. CD, cow dung; PL, poultry litter; SS, sewage sludge; LGM, groundnut + millet from Saint-Louis; BG, groundnut from the southern groundnut basin; BR, rice from the southern groundnut basin; TOC, total organic carbon; TON, total organic nitrogen; Assim. P, assimilable phosphorus; WC, water content; CEC, cation exchange capacity; n.d., not determined; DM, dry matter.

### 2.2. Organic Waste Products

Agricultural (PL et CD) and urban (SS) OWPs were sampled in the Dakar region (Senegal). Fresh cow dung and dry poultry litter (droppings mixed with groundnut shells) were sampled at Sangalkam in a cattle paddocking site and a chicken coop, respectively. Sewage sludge was sampled at the Cambérène wastewater treatment plant in Dakar (processed by methanation and total open bed drying). The main OWP chemical properties are listed in Table 1.

### 2.3. Local Beneficial Microorganisms

BMs were derived from forest litter collected in two Senegalese regions (Saint-Louis and the southern groundnut basin). For a 200 L drum of solid matter, 23 kg of raw forest litter (dead leaves and branches with a high microorganism content) was thoroughly mixed with 10 kg of sugarcane molasses (source of quick-acting carbohydrates), 5 kg of yoghurt (source of lactic acid bacteria), 46 kg of rice bran, millet husks or groundnut shells (source of carbon), and 63.8 L of demineralized water. The resulting product was anaerobically fermented for 1 month at ambient temperature. The main chemical properties of the three BMs used (LGM, BG, and BR) are presented in Table 1.

To determine the overall diversity and abundance of BMs, total genomic DNA was extracted from 25 g of fresh sample (after 1 month of fermentation), using the FastDNA™ SPIN kit (MP Biomedicals, CA, Irvine (California), USA), with modifications to the manufacturer's instructions [58]. The quality and concentration of the extracted DNA were verified after migration on 1.5% agarose gel electrophoresis. Amplification and high-throughput sequencing of bacterial and fungi were performed by the ADNID company (http://www.adnid.fr, accessed on 15 December 2022, Montferrier sur Lez, France) with a MiSeq Illumina system by targeting 16S rRNA gene with the 515F/806R primers set and ITS gene, using the ITS3F-ITS4R primers. Sequence data processing was conducted at ADNID (http://www.adnid.fr, accessed on 15 December 2022, Montferrier sur Lez, France;). Briefly, raw Illumina MiSeq paired-end reads were assembled, and sequences were demultiplexed and formatted for processing using a Python script (http://drive5.com/usearch/manual/uparse_pipeline.html, accessed on 15 December 2022). Sequences were then separately quality-filtered and clustered into operational taxonomic units (OTUs) at 3% divergence (97% similarity), using the UPARSE algorithm [59]. The taxonomic affiliation of each OTU was obtained using BLASTn against a curated database derived from GreenGenes [60] and SILVA databases (https://www.arb-silva.de/, accessed on 15 December 2022).

### 2.4. Incubation Experiment

A 28-day incubation was performed under temperature- and humidity-controlled laboratory conditions by adapting the AFNOR-FDU44-163 [61] standard to our local conditions, using the sampled soil, the three OWPs (PL, CD, and SS), and the three BMs (LGM, BG, and BR). With these different resources, a total of 16 microcosms were obtained, as listed in Table 2. Briefly, the soil was pre-moistened with demineralized water to 80% of its maximum water retention capacity. One-week pre-incubation at 28 °C was carried out to reactivate the microbiological activity and limit the mineralization flush that usually occurs when rewetting dry soil during incubation [62,63]. Soil alone was used as an experimental control. The OWP input rates were 33.3 g kg$^{-1}$ dry soil (1 g dry equivalent/30 g of soil) [64], and the BM input rates were 50 g kg$^{-1}$ dry soil (1.5 g dry equivalent/30 g of soil), representing inputs ranging from 7.91 to 10.65 gOC kg$^{-1}$ dry soil for OWP and from 11.92 to 17.12 gOC kg$^{-1}$ dry soil for BM, respectively. The different microcosms (control soil, soil + OWP, soil + BM, and soil + OWP + BM) were prepared in triplicate. As gas chromatography measurements of C-CO$_2$ are nondestructive, 48 microcosms were prepared to monitor the carbon mineralization dynamics. Because the extraction (with 40 mL KCl 1 N per 10 g soil) and measurement of mineral nitrogen in the soil solution (with the SEAL Technicon colorimetric analysis system) were destructive processes, triplicate samples were prepared for each measurement date (i.e., 288 microcosms for monitoring the nitrogen mineralization dynamics (Supplementary Image S1)). The microcosms were placed in 2 L jars containing 5 mL of demineralized water to ensure that the air remained humid. The jars were incubated in a thermostatically controlled oven at 28 °C for 28 days. The microcosm humidity was checked by weighing twice a week throughout the incubation period and readjusted if necessary.

**Table 2.** Microcosm names.

| Names | | Soil | OWP | | | BM | | |
|---|---|---|---|---|---|---|---|---|
| **Generic** | **Specific** | | **CD** | **PL** | **SS** | **LGM** | **BG** | **BR** |
| Control | Control | x | | | | | | |
| S-OWPs | S-CD | x | x | | | | | |
| | S-PL | x | | x | | | | |
| | S-SS | x | | | x | | | |
| S-BMs | S-LGM | x | | | | x | | |
| | S-BG | x | | | | | x | |
| | S-BR | x | | | | | | x |
| S-OWPs-BMs | S-CD-LGM | x | x | | | x | | |
| | S-CD-BG | x | x | | | | x | |
| | S-CD-BR | x | x | | | | | x |
| | S-PL-LGM | x | | x | | x | | |
| | S-PL-BG | x | | x | | | x | |
| | S-PL-BR | x | | x | | | | x |
| | S-SS-LGM | x | | | x | x | | |
| | S-SS-BG | x | | | x | | x | |
| | S-SS-BR | x | | | x | | | x |

OWPs, organic waste products; BMs, local beneficial microorganisms; S, soil; CD, cow dung; PL, poultry litter; SS, sewage sludge; LGM, groundnut + millet from Saint-Louis; BG, groundnut from the southern groundnut basin; BR, rice from the southern groundnut basin.

### 2.5. Carbon Mineralization Kinetics

Carbon mineralization kinetics were determined based on $C-CO_2$ measurements obtained by gas chromatography (Agilent 490 Micro Gas Chromatograph, Santa Clara, CA, USA) with EZChrom A.04.10 software (Santa Clara, CA, USA) (Supplementary Image S2). Cumulated $C-CO_2$ emissions in the microcosms were measured at 0, 1, 3, 7, 14, and 28 days of incubation.

Cumulated $C-CO_2$ emissions (expressed in $mgC-CO_2$) in the different microcosms (control soil, soil + OWP, soil + BM, and soil + OWP + BM) were calculated according to the AFNOR-FDU44-163 [61] standard.

In the S-OWP and S-BM microcosms, cumulated $C-CO_2$ emissions were calculated and normalized according to the OC level, as follows:

$$C_i^{\%} = \frac{oC_{si} - oC_s}{OC_i} \times 100$$

where $C_i^{\%}$ is the cumulated quantity of carbon (in $gC-CO_2$ $100\ g^{-1}$ of OC) emitted by the ith OWP or BM; $oC_{si}$ is the cumulated observed quantity of carbon (in $gC-CO_2$) emitted by the microcosm containing soil + ith OWP or BM; $oC_s$ is the cumulated observed quantity of carbon (in $gC-CO_2$) emitted by the microcosm containing soil alone; and $OC_i$ is the quantity of organic carbon (in gOC) provided by the ith OWP or BM.

The cumulated observed quantities of $C-CO_2$ emitted by the OWP-BM combinations were determined as follows:

$$oC_{ij} = oC_{sij} - oC_s$$

where $oC_{ij}$ is the cumulated observed quantity of carbon (in $mgC-CO_2$) emitted by the ith OWP and jth BM combination; $oC_{sij}$ is the cumulated observed quantity of carbon (in $mgC-CO_2$) emitted by the microcosm containing soil + ith OWP + jth BM; and $oC_s$ is the cumulated observed quantity of carbon (in $mgC-CO_2$) emitted by the microcosm containing soil alone.

The cumulated quantities of C-CO$_2$ calculated for the OWP-BM combinations were determined as follows:

$$cC_{ij} = (oC_{si} - oC_s) + (oC_{sj} - oC_s)$$

where $cC_{ij}$ is the cumulated calculated quantity of carbon (in mgC-CO$_2$) emitted by the ith OWP and jth BM combination; $oC_{si}$ is the cumulated observed quantity of carbon (in mgC-CO$_2$) emitted by the microcosm containing soil + ith OWP; $oC_{sj}$ is the cumulated observed quantity of carbon (in mgC-CO$_2$) emitted by the microcosm containing soil + jth BM; and $oC_s$ is the cumulated observed quantity of carbon (in mgC-CO$_2$) emitted by the microcosm containing soil alone.

In the S-OWP-BM microcosms, cumulated quantities of C-CO$_2$ expressed as gC-CO$_2$ 100 g$^{-1}$ OC of the applied OWP were calculated based on the assumption (Hypothesis 1) that the carbon in the applied BM was not mineralized because its C: N ration was higher than that in the applied OWP. This was calculated as follows:

$$C_{ij}^\% = \frac{oC_{sij} - oC_s}{OC_i} \times 100$$

where $C_{ij}^\%$ is the cumulated quantity of carbon (in gC-CO$_2$ 100 g$^{-1}$ of OC) emitted by the ith OWP and jth BM combination; $oC_{sij}$ is the cumulated observed quantity of carbon (in gC-CO$_2$) emitted by the microcosm containing soil + ith OWP + jth BM. $oC_s$ is the cumulated observed quantity of carbon (in gC-CO$_2$) emitted by the microcosm containing soil alone; and $OC_i$ is the quantity of organic carbon (in gOC) provided by the ith OWP.

We also put forward a second hypothesis (Hypothesis 2) for the cumulated normalized C-CO$_2$ quantity calculations based on the assumption that, in S-OWP-BM microcosms, carbon in the applied BM is mineralized at the same rate as in the S-BM microcosms. Our calculations under this hypothesis are outlined in the Supplementary Materials (SMs).

*2.6. Dynamics of Organic Nitrogen Transformation into Its Main Mineral Forms*

N-NO$_3^-$ and N-NH$_4^+$ quantities were measured after 1 M KCl extraction (with 40 mL KCl 1 N per 10 g soil), and they were determined using the SEAL Technicon colorimetric analysis system (Supplementary Image S3). Nitrogen mineralization kinetics were determined from measurements obtained at 0, 1, 3, 7, 14, and 28 days of incubation. Nmin quantities were obtained by adding the N-NH$_4^+$ and N-NO$_3^-$ quantities.

Nmin quantities (expressed in mgNmin) in the different microcosms (control soil, soil + OWP, soil + BM, and soil + OWP + BM) were calculated according to the AFNOR-FDU44-163 [61] standard.

In the S-OWP and S-BM microcosms, the net Nmin quantities calculated and normalized according to the organic nitrogen (ON) quantity were determined as follows:

$$N_i^\% = \frac{(oN_{si} - oN_s)}{ON_i} \times 100$$

where $N_i^\%$ is the net nitrogen quantity (in gNmin 100 g$^{-1}$ of ON) emitted by the ith OWP or BM; $oN_{si}$ is the observed quantity of nitrogen (in gNmin) emitted by the microcosm containing soil + ith OWP or BM; $oN_s$ is the observed quantity of nitrogen (in gNmin) emitted by the microcosm containing soil alone; and $ON_i$ is the quantity of organic nitrogen (in gON) provided by the ith OWP or BM.

The net observed Nmin quantities of the OWP-BM combinations were determined as follows:

$$oN_{ij} = oN_{sij} - oN_s$$

where $oN_{ij}$ is the net observed nitrogen quantity (in mgNmin) emitted by the ith OWP and jth BM combination; $oN_{sij}$ is the observed quantity of nitrogen (in mgNmin) emitted by

the microcosm containing soil + ith OWP + jth BM; and $oN_s$ is the observed quantity of nitrogen (in mgNmin) emitted by the microcosm containing soil alone.

The net calculated Nmin quantities of the OWP-BM combinations were determined as follows:

$$cN_{ij} = (oN_{si} - oN_s) + (oN_{sj} - oN_s)$$

where $cN_{ij}$ is the net calculated nitrogen quantity (in mgNmin) emitted by the ith OWP and jth BM combination; $oN_{si}$ is the observed quantity of nitrogen (in mgNmin) emitted by the microcosm containing soil + ith OWP; $oN_{sj}$ is the observed quantity of nitrogen (in mgNmin) emitted by the microcosm containing soil + jth BM; and $oN_s$ is the observed quantity of nitrogen (in mgNmin) emitted by the microcosm containing soil alone.

In the S-OWP-BM microcosms, the net Nmin quantities calculated and normalized according to the organic N input were determined as follows:

$$N_{ij}^{\%} = \frac{(oN_{sij} - oN_s)}{(ON_i + ON_j)} \times 100$$

where $N_{ij}^{\%}$ is the net nitrogen quantity (in gNmin 100 g$^{-1}$ of ON) emitted by the ith OWP and jth BM combination; $oN_{sij}$ is the observed nitrogen quantity (in gNmin) emitted by the microcosm containing soil + ith OWP + jth BM; $oN_s$ is the observed quantity of nitrogen (in mgNmin) emitted by the microcosm containing soil alone; $ON_i$ is the organic nitrogen quantity (in gON) provided by the ith OWP; and $ON_j$ is the organic nitrogen quantity (in gON) provided by the jth BM.

### 2.7. Statistical Analysis

Statistical analyses were performed with the R software package (version 4.2.1). The data normality was previously assessed by the graphical display (boxplot and QQ-plot) and the Shapiro–Wilk test. The variance homogeneity was checked using Levene's test. The OWP and BM effects were assessed via a one-way analysis of variance (ANOVA). Means were compared by the Newman–Keuls test at the 5% threshold. The Kruskal–Wallis test was applied for abnormally distributed data. The Wilcoxon test at the 5% threshold was used to compare observed and calculated means of the OWP-BM combinations to assess interactions between OWP and BM.

All the individual numeric values of the means, standard deviations, and *p*-values are provided in detail in the Supplementary Materials Tables S1a–S6c.

### 3. Results

#### 3.1. BM Microbial Composition

The three assessed BMs differed in their microbial composition (Table 3). Regarding bacteria, at the phylum level, LGM and BG were dominated by Firmicutes (83.9% and 68.14%, respectively) and Proteobacteria (8.59% and 30.93%, respectively), whereas Proteobacteria (45.32%), Firmicutes (26.30%), and Bacteroidota (25.41%) were the most dominant phyla in BR.

LGM mainly contained Firmicutes (83.85%), dominated by *Lactobacillus* (62.5%) and *Bacillus* (16.45%), followed by Actinobacteria (6.39%) and Proteobacteria (5.65%). BG had fewer Firmicutes (57.16%) than LGM yet still had a high proportion of Proteobacteria (30.27%). BR was dominated by Proteobacteria (38.79%), Bacteroidota (25.41%), and Firmicutes (23.93%, but their relative abundance was lower than in LGM and BG).

Regarding fungi, LGM had the highest fungal community diversity compared to BG and BR. It was dominated by the largest and most diverse classes of Ascomycetes, i.e., Dothideomycetes (63%), Sordariomycetes (23.8%), and Saccharomycetes (11.1%), whereas the BG and BR compositions were predominantly represented by Saccharomycetes (99.66% and 88.19%, respectively). BG and BR had the lowest relative abundance of Dothideomycetes (0.2 and 7%, respectively) compared to LGM.

LGM was dominated by *Cladosporium* (37.80%), *Myrothecium* (18.16%), *Mycosphaerella* (13.9%), *Yueomyces* (11.14%), and *Colletotrichum* (5%), while BG and BR were mainly represented by *Yueomyces*, especially *Yueomyces sinensis*, while having higher relative abundances (99.66% and 87.96%, respectively).

**Table 3.** Diversity and abundance of BMs.

| | Groups | Genus | LGM | BG | BR |
|---|---|---|---|---|---|
| | | *Acetobacter* | 0.01 | 28.45 | 0.81 |
| | | *Acinetobacter* | 0.01 | 0.01 | 10.79 |
| | | *Pseudomonas* | 0.01 | 0.01 | 3.09 |
| | | *Enterobacter* | 0.07 | 0.34 | 9.19 |
| | | *Pantoea* | 0.30 | 0.13 | 0.07 |
| | | *Rhizobium* | 0.81 | 0.05 | 0.22 |
| | Proteobacteria | *Methylobacterium* | 0.04 | 0.01 | 0.003 |
| | | *Stenotrophomonas* | 0.01 | 0.01 | 17.82 |
| | | *Burkholderia* | 0.01 | 0.01 | 0.001 |
| | | *Bradyrhizobium* | 0.01 | 0.03 | 0 |
| | | *Mesorhizobium* | 0.20 | 0 | 0 |
| Bacteria | | *Devosia* | 0.90 | 0.03 | 0.01 |
| (% OTU) | | *Pseudoaminobacter* | 0.01 | 0 | 0 |
| | Actinobacteria | *Mycobacterium* | 0.60 | 0.01 | 0.02 |
| | | *Lactobacillus* | 61.51 | 40.00 | 6.95 |
| | | *Bacillus* | 9.63 | 5.35 | 1.81 |
| | | *Weissella* | 1.87 | 5.08 | 0.28 |
| | | *Paenibacillus* | 1.17 | 0.27 | 0.47 |
| | Firmicutes | *Clostridium* | 0.01 | 8.01 | 0.28 |
| | | *Lysinibacillus* | 1.11 | 0.83 | 4.41 |
| | | *Rummeliibacillus* | 0.07 | 2.92 | 1.38 |
| | | *Pediococcus* | 1.21 | 0.74 | 0.03 |
| | | *Virgibacillus* | 1.62 | 0.01 | 0.003 |
| | Bacteroidota | *Sphingobacterium* | 0.01 | 0.02 | 23.70 |
| | | *Chryseobacterium* | 0 | 0 | 5.29 |
| | | *Neoascochyta* | 0.001 | 33.75 | 35.18 |
| Fungi | | *Ascochyta* | 2.85 | 3.59 | 10.98 |
| (% OTU) | Ascomycota | *Myrothecium* | 41.84 | 2.13 | 1.77 |
| | | *Cladosporium* | 24.82 | 0.01 | 0.024 |
| | | *Colletotrichum* | 10.67 | 0.05 | 0.56 |
| | Basidiomycota | *Yueomyces* | 3.36 | 2.11 | 0.23 |

OTU, operational taxonomic unit; LGM, groundnut + millet from Saint-Louis; BG, groundnut from the southern groundnut basin; BR, rice from the southern groundnut basin.

### 3.2. Impact of OWP or BM on Carbon Mineralization in Microcosms

The OWP input altered the carbon mineralization kinetics in the microcosms. There was a significant increase ($p < 0.0001$) in cumulated $C\text{-}CO_2$ quantities in S-OWP microcosms (Figure 1A) compared to the control throughout incubation. The cumulated $C\text{-}CO_2$ at 28 days ranged from $18.41 \pm 0.42$ to $168.63 \pm 4.02$ mg for the S-OWP microcosms but was only $3.33 \pm 0.15$ mg in the control. This corresponds to increases in cumulated $C\text{-}CO_2$ quantities ranging from 5.5 to 50-fold in S-OWP microcosms compared to that in the control.

The carbon mineralization kinetics differed according to the type of OWP. The S-OWP microcosms were ranked in the order of S-PL > S-CD > S-SS according to their cumulated $C\text{-}CO_2$ quantity ($p < 0.0001$). The cumulated $C\text{-}CO_2$ at 28 days was $61.09 \pm 1.48$ g 100 g$^{-1}$ for S-PL, $23.40 \pm 1.04$ g 100 g$^{-1}$ for S-CD, and $6.43 \pm 0.18$ g 100 g$^{-1}$ for S-SS (Figure 1C). At the end of incubation, the S-PL microcosm contained 2.6- and 9.5-fold more $C\text{-}CO_2$ than the S-CD and S-SS microcosms, respectively.

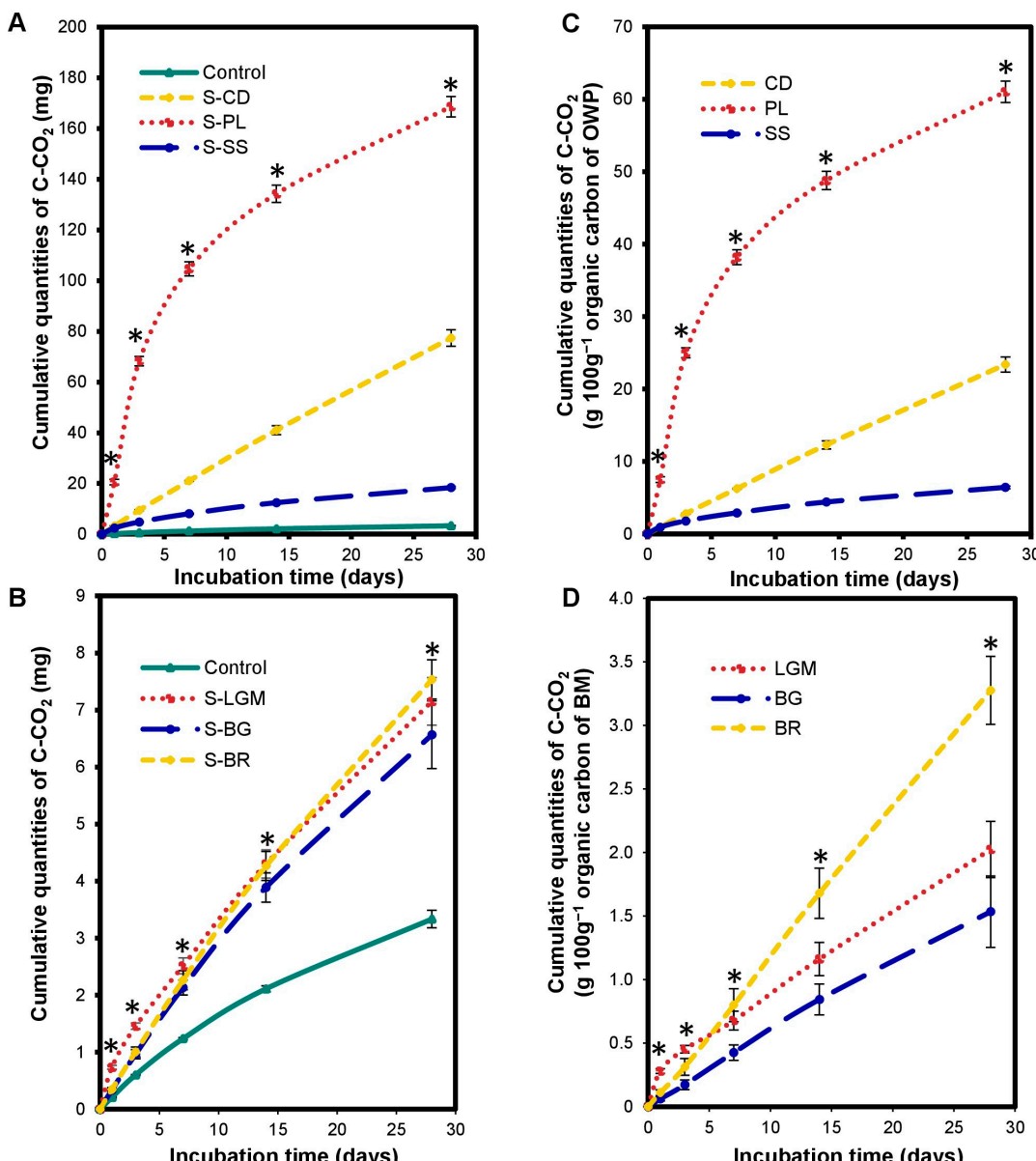

**Figure 1.** Variations in cumulated quantities of C-CO$_2$ in the microcosms expressed in mgC-CO$_2$ (**A**,**B**) and in gC-CO$_2$ 100 g$^{-1}$ of organic carbon from OWP (**C**) or BM (**D**). OWPs, organic waste products; BMs, local beneficial microorganisms; S, soil; CD, cow dung; PL, poultry litter; SS, sewage sludge; LGM, groundnut + millet from Saint-Louis; BG, groundnut from the southern groundnut basin; BR, rice from the southern groundnut basin. Error bars correspond to the standard deviation of three replicates. An asterisk (*) shows a significant difference between values within an incubation time at $p < 0.05$. Specific statistical data related to this figure are reported in Supplementary Table S1a–d.

The BM input altered the carbon mineralization kinetics in the microcosms. There was a significant increase ($p < 0.0001$) in cumulated C-CO$_2$ quantities in the S-BM microcosms (Figure 1B) compared to the control throughout incubation. Cumulated C-CO$_2$ quantities at 28 days ranged from $6.57 \pm 0.59$ to $7.54 \pm 0.34$ mg in the S-BM microcosms but were only $3.33 \pm 0.15$ mg in the control. This corresponds to increases in cumulated C-CO$_2$ quantities ranging from 1.9- to 2.2-fold in the S-BM microcosms compared to the control. Note that the cumulated C-CO$_2$ quantities in the S-BM microcosms were 2–60-fold lower than those in the S-OWP microcosms at 28 days, although the OC inputs in the BMs were 1.11-to-2.16-fold higher than those in the OWPs. At the end of incubation, the S-BR microcosm contained 1.6- and 2.1-fold more C-CO$_2$ than the S-LGM and S-BG microcosms, respectively.

The S-BM microcosms were ranked in the order of S-BR > S-LGM $\geq$ S-BG according to their cumulated C-CO$_2$ quantities ($p < 0.001$; Figure 1D). The cumulated C-CO$_2$ quantities at 28 days were 3.28 $\pm$ 0.27 g.100 g$^{-1}$ for S-BR, 2.02 $\pm$ 0.22 g.100 g$^{-1}$ for S-LGM, and 1.53 $\pm$ 0.28 g.100 g$^{-1}$ for S-BG.

### 3.3. Impact of the Interaction between OWP and BM on Microcosm Carbon Mineralization

A positive and significant impact of the interaction between OWP and BM on microcosm carbon mineralization was noted (Figure 2). Cumulated C-CO$_2$ quantities measured for the OWP-BM combinations were higher than those calculated ($p < 0.05$), except for SS-LGM. The CD-BG, CD-LGM, and PL-LGM combinations had the highest carbon mineralization rates measured at 28 days (compared to the calculated carbon mineralization rates), i.e., 22%, 18%, and 10%, respectively ($p < 0.0001$).

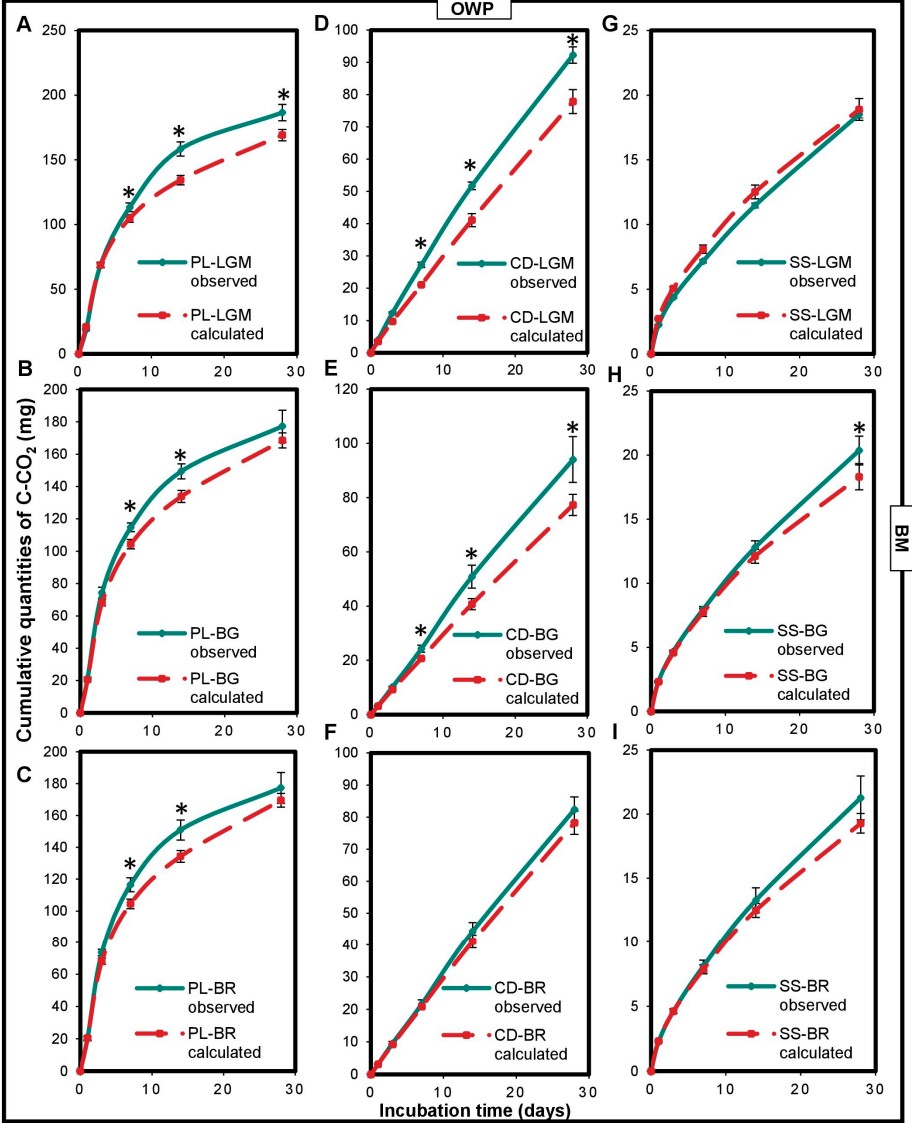

**Figure 2.** Variations in cumulated quantities of C-CO$_2$ expressed in the observed mgC-CO$_2$ and in calculated mgC-CO$_2$ of the OWP-BM combinations (**A–I**). OWPs, organic waste products; BMs, local beneficial microorganisms; S, soil; CD, cow dung; PL, poultry litter; SS, sewage sludge; LGM, groundnut + millet from Saint-Louis; BG, groundnut from the southern groundnut basin; BR, rice from the southern groundnut basin. Error bars correspond to the standard deviation of three replicates. An asterisk (*) shows a significant difference between values within an incubation time at $p < 0.05$. Specific statistical data related to this figure are reported in Supplementary Table S2a–e and S2h.

### 3.4. Impact of BM on OWP Carbon Mineralization

The BM input induced excessive carbon mineralization in OWPs compared to OWPs alone (Figure 3). The cumulated $C-CO_2$ quantities of OWP-BM combinations are higher than those of OWP alone during incubation. The excess mineralization rate depended on (i) the BM type (Figure 3C)—for instance, SS carbon was mineralized excessively by 23% by LGM, and by 41% by BR, compared to SS alone at 28 days ($p < 0.001$); and (ii) the incubation time (Figure 3A)—for instance, PL carbon was mineralized excessively by LGM by 13% at 7 and 28 days and by 20% at 14 days, compared to PL alone ($p < 0.05$). However, the excess mineralization rate of OWP carbon induced by BM compared to OWP alone was higher between 14 and 28 days. The excess mineralization of PL and CD was more marked with LGM (Figure 3A,B), while that of SS carbon was more marked with BR (Figure 3C).

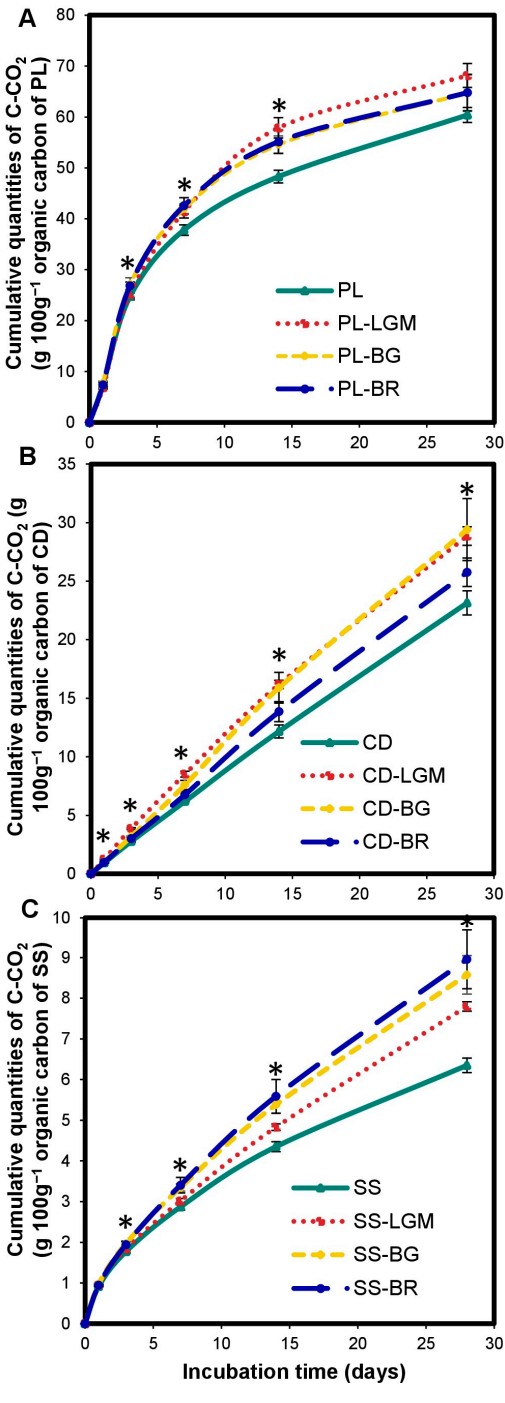

**Figure 3.** Variations in cumulated $C-CO_2$ quantities of OWP expressed in $gC-CO_2$ $100$ $g^{-1}$ of organic

carbon of OWP (Hypothesis 1, **A**–**C**). OWPs, organic waste products; BMs, local beneficial microorganisms; S, soil; CD, cow dung; PL, poultry litter; SS, sewage sludge; LGM, groundnut + millet from Saint-Louis; BG, groundnut from the southern groundnut basin; BR, rice from the southern groundnut basin. Error bars correspond to the standard deviation of three replicates. An asterisk (*) shows a significant difference between values within an incubation time at $p < 0.05$. Specific statistical data related to this figure are reported in Supplementary Table S3a–c.

Excess mineralization of OWP carbon by BMs (compared to OWP alone) was also noted under the Hypothesis 2 (Supplementary Figure S1). However, the excess mineralization rates obtained were lower than those under the Hypothesis 1 (Figure 3). For instance, CD carbon was excessively mineralized by BG by 26% under the first hypothesis (Figure 3B) but only by 22% under the second hypothesis (Supplementary Figure S1B). However, the BM input induced excessive mineralization of OWP carbon (compared to OWP alone), regardless of the hypothesis.

### 3.5. Impact of OWP or BM on Microcosm Nitrogen Mineralization

The OWP input altered ($p < 0.0001$) the nitrogen mineralization kinetics in the microcosms relative to the control (Figure 4A). The increase or reduction of Nmin quantities by OWP depends on the incubation time. The S-OWP microcosms were ranked as follows: S-SS > control > S-CD from 1 to 28 days; S-PL > control from 1 to 7 days; and Control > S-PL from 14 to 28 days.

The modification of nitrogen mineralization kinetics by the addition of OWP depends on the type of OWP. The S-OWP microcosms were ranked in the order of S-SS > S-PL > S-CD, between 7 and 28 days (Figure 4C), according to their net Nmin quantities ($p < 0.05$). The net Nmin quantities at 28 days were $10.37 \pm 0.12$ g.100 g$^{-1}$ for S-SS, $-0.60 \pm 0.07$ g.100 g$^{-1}$ for S-PL, and $-7.96 \pm 0.04$ g.100 g$^{-1}$ for S-CD.

The BM input altered ($p < 0.0001$) the microcosm N mineralization kinetics. The increase or reduction of Nmin quantities by BM depends on the incubation time and type of BM. When comparing the Nmin quantities in the S-BM microcosms to the control (Figure 4B), they were ranked as follows: S-LGM > control from 1 to 28 days; control > S-BR from 7 to 28 days; S-BG > control at 3 and 14 days; and control > S-BG at 1, 7, and 28 days. When comparing the net Nmin quantities in the S-BM microcosms with each other (Figure 4D), they were ranked as follows: S-LGM > S-BG > S-BR ($p < 0.05$). The net Nmin quantities at 28 days were $25.72 \pm 1.39$ g.100 g$^{-1}$ for S-LGM, $-9.11 \pm 0.24$ g.100 g$^{-1}$ for S-BG, and $-25.08 \pm 0.93$ g.100 g$^{-1}$ for S-BR.

### 3.6. Impact of the Interaction between OWP and BM on Microcosm Nitrogen Mineralization

An interaction between OWP and BM impacting nitrogen mineralization in microcosms was observed (Figure 5). This impact was significantly ($p < 0,05$) positive (measured net Nmin quantities higher than calculated net Nmin quantities) or negative (calculated net Nmin quantities higher than measured net Nmin quantities), depending on the type of OWP-BM combination and the incubation time. PL had a positive interaction with each of the BMs. Indeed, except for at the beginning of incubation, the measured net Nmin quantities of the PL-LGM, PL-BG, and PL-BR combinations were higher than those calculated (Figure 5A–C). CD had a significantly ($p < 0.0001$) negative interaction with LGM (between 1 and 28 days) or BG (between 3 and 14 days) (Figure 5D,E) but a positive interaction with BR (Figure 5F). SS had a negative interaction with LGM, BG, or BR between 3 and 14 days but a positive interaction with BG or BR at 28 days (Figure 5G–I). The measured net Nmin quantities of the PL-LGM, PL-BG, and PL-BR combinations were, respectively, 14-, 6.5-, and 5.4-fold higher than their calculated net Nmin quantities at 28 days ($p < 0.0001$).

### 3.7. Impact of BM on OWP Nitrogen Mineralization

The OWP-BM combinations altered the microcosm N mineralization kinetics compared to OWP alone (Figure 6). The significant ($p < 0.05$) increase or reduction in net

Nmin quantities depended on (i) the type of OWP—for instance, the net Nmin quantity of PL-LGM was significantly ($p < 0.0001$) higher than that of PL between 14 and 28 days (Figure 6A), whereas the net Nmin quantity of CD-LGM was significantly ($p < 0.0001$) lower than that of CD during the same incubation period (Figure 6B); (ii) BM type—for instance, by day 14 (Figure 6B), the nitrogen mineralization rate of CD-BR was 16.4-fold higher than that of CD, but the nitrogen mineralization rate of CD-LGM was 2.6-fold lower than that of CD; and (iii) incubation time—for instance, the N mineralization rate of SS was 1.8-fold higher than that of SS-BG at 14 days (Figure 6C), while it was 1.7-fold lower than that of SS-BG at 28 days ($p < 0.0001$). The net Nmin quantities at 28 days were $16.99 \pm 0.08$ for PL-LGM and $17.19 \pm 0.09$ for SS-BG.

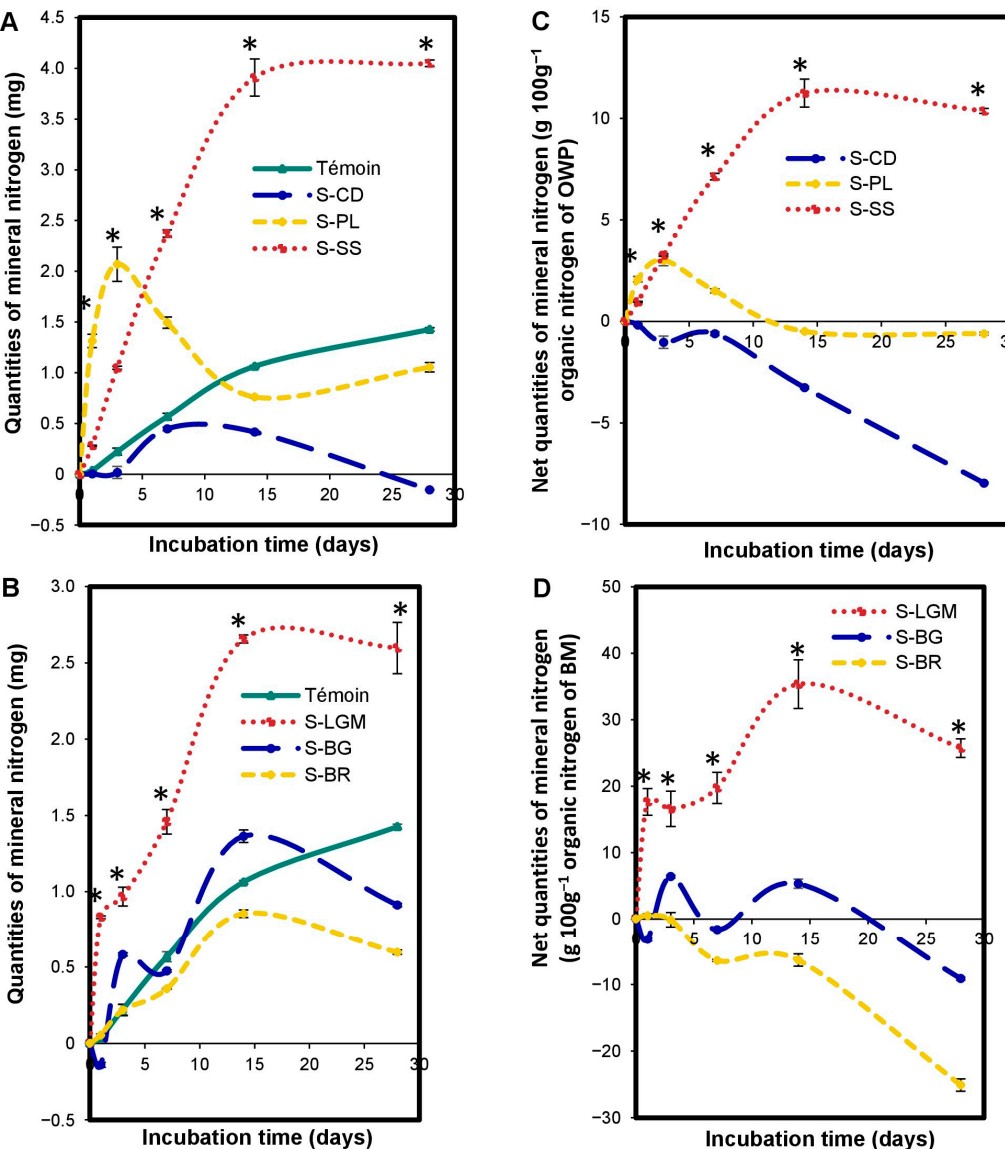

**Figure 4.** Variations in microcosm mineral N quantities expressed in mgNmin (**A,B**), and net microcosm mineral N quantities expressed in gNmin 100 g$^{-1}$ of organic N from OWP (**C**) or BM (**D**). OWPs, organic waste products; BMs, local beneficial microorganisms; S, soil; CD, cow dung; PL, poultry litter; SS, sewage sludge; LGM, groundnut + millet from Saint-Louis; BG, groundnut from the southern groundnut basin; BR, rice from the southern groundnut basin. Error bars correspond to the standard deviation of three replicates. An asterisk (*) shows a significant difference between values within an incubation time at $p < 0.05$. Specific statistical data related to this figure are reported in Supplementary Table S4a–d.

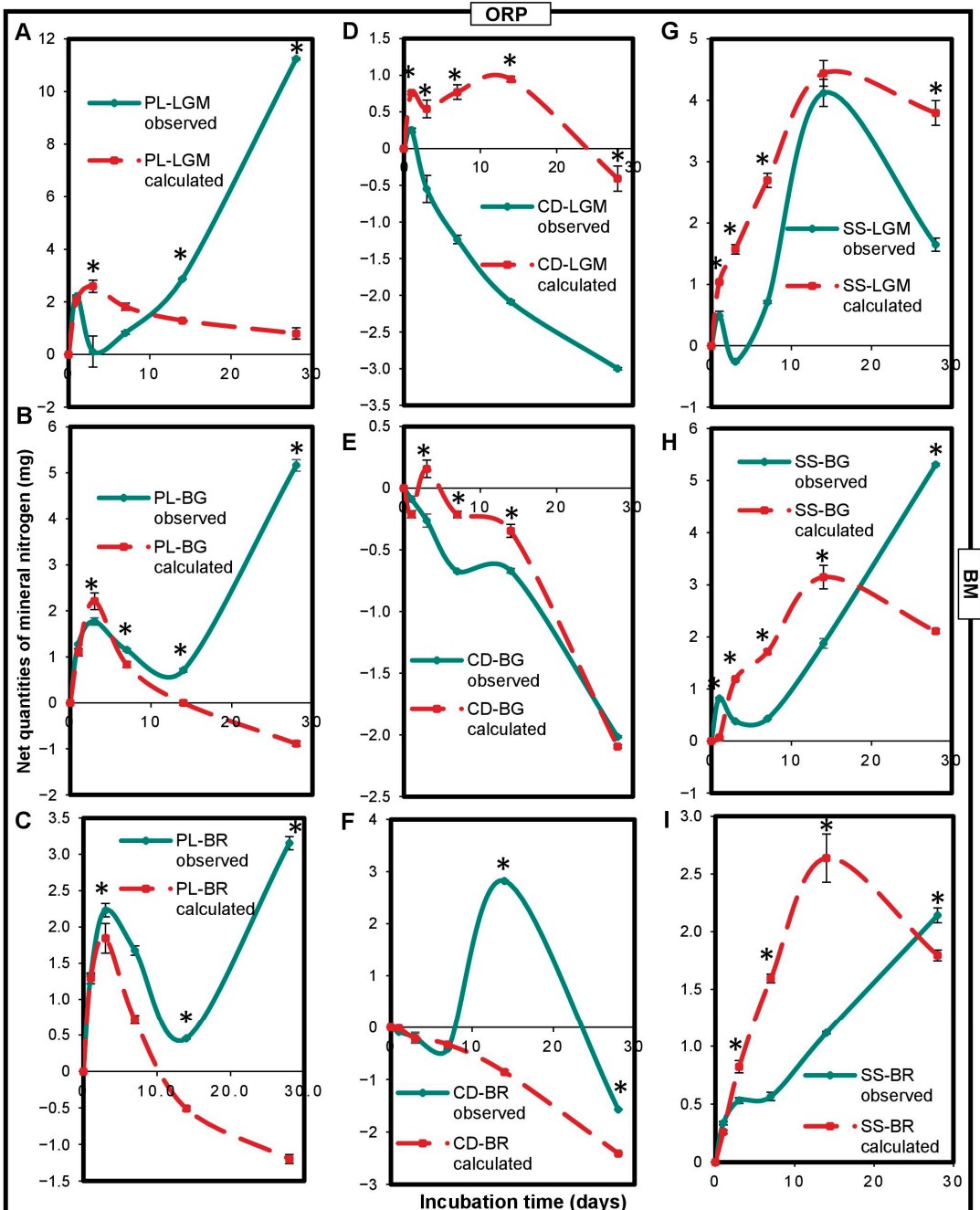

**Figure 5.** Variations in net mineral nitrogen quantities expressed in observed mgNmin and calculated mgNmin of OWP-BM combinations (**A–I**). Nmin, mineral nitrogen; OWPs, organic waste products; BMs, local beneficial microorganisms; S, soil; CD, cow dung; PL, poultry litter; SS, sewage sludge; LGM, groundnut + millet from Saint-Louis; BG, groundnut from the southern groundnut basin; BR, rice from the southern groundnut basin. Error bars correspond to the standard deviation of three replicates. An asterisk (*) shows a significant difference between values within an incubation time at $p < 0.05$. Specific statistical data related to this figure are reported in Supplementary Table S5a–i.

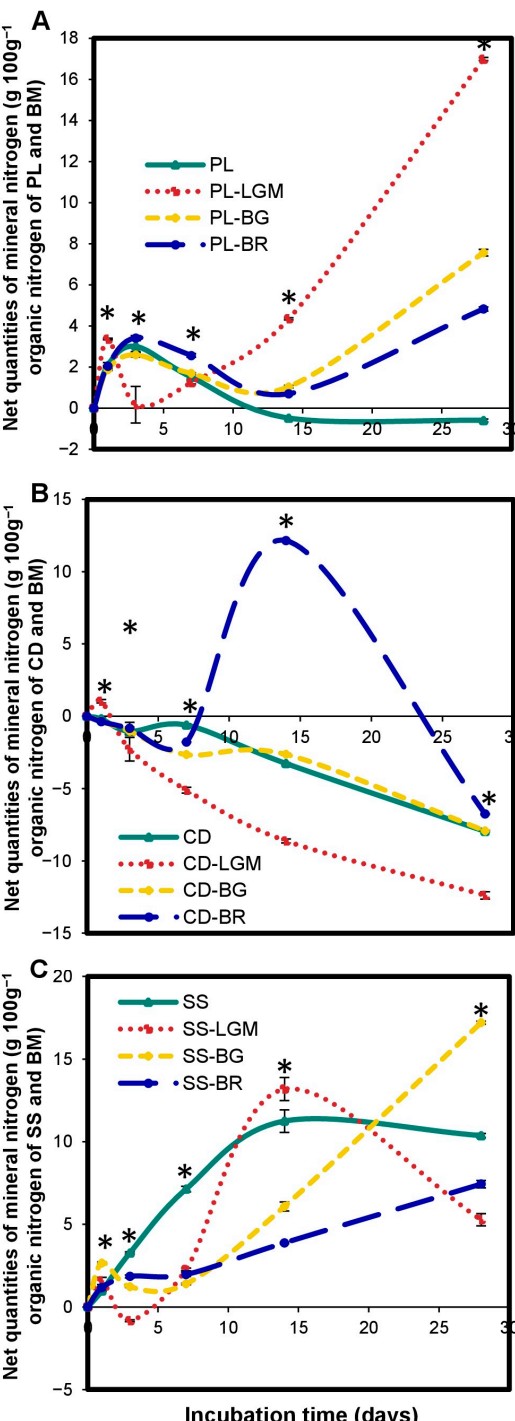

**Figure 6.** Variations in net mineral nitrogen quantities in microcosms expressed in gNmin 100 g$^{-1}$ of organic nitrogen from OWP and BM (**A–C**). OWPs, organic waste products; BMs, local beneficial microorganisms; CD, cow dung; PL, poultry litter; SS, sewage sludge; LGM, groundnut + millet from Saint-Louis; BG, groundnut from the southern groundnut basin; BR, rice from the southern groundnut basin. Error bars correspond to the standard deviation of three replicates. An asterisk (*) shows a significant difference between values within an incubation time at $p < 0.05$. Specific statistical data related to this figure are reported in Supplementary Table S6a–c.

## 4. Discussion

### 4.1. Impact of OWP or BM on Carbon Mineralization in Microcosms

The observed increase in microcosm carbon mineralization resulting from OWP input in our study was consistent with previous findings under a diverse range of settings, soils, OWP types, and dosages [65–68]. For instance, Dossa et al. [69] observed a significant 15-fold increase in $C\text{-}CO_2$ quantities in a sandy loam soil amended with crop residues or beef manure, compared with the soil alone, during a 120-day incubation. This significant OWP impact on microcosm carbon mineralization is not very dependent on the soil type [32,70] but rather on the nature and composition of the applied OWP.

According to their impact on carbon mineralization, the OWPs used in our study were ranked in the order of PL > CD > SS, whereas they were ranked in the order of CD > PL > SS based on their C input quantities (CD = 10.65, PL = 9.11, and SS = 7.91 $\text{gOC kg}^{-1}$ dry soil). Moreover, the OWP ranking order, according to their carbon mineralization effect, was the same as that established according to their DOC ranking (PL > CD > SS) (Supplementary Figure S2A). This suggests that the impact of an OWP on microcosm carbon mineralization may depend on its DOC. Actually, the high concentration of readily biodegradable organic carbon introduced with the application of an organic product stimulates microcosm microbial activity [34,35,71]. Table 1 shows that the C: N ratio of PL was lower than that of SS. However, poultry manure—with a higher C: N ratio than sewage sludge—had a higher $C\text{-}CO_2$ quantity than sewage sludge in a previous study of Miyittah and Inubushi [72], indicating that carbon mineralization is more related to the organic matter decomposability than to the C: N ratio [25,73,74].

The $C\text{-}CO_2$ quantities measured with our OWPs at 28 days of incubation differed from previously reported results regarding quantities measured with the same types of OWPs at 28 days. For instance, the $C\text{-}CO_2$ quantity measured for sewage sludge at 28 days (6 $\text{g.100 g}^{-1}$) was lower than that measured with sewage sludge at 28 days ($\approx$31 $\text{g.100 g}^{-1}$) by [75]. This might be explained by the atypical nature of our OWPs. Indeed, the sewage sludge treatment process used in our study included methanation and total open bed drying, which probably consumed carbon and reduced the sludge-borne microbial community. Furthermore, the $C\text{-}CO_2$ quantity measured for poultry litter at 28 days (61 $\text{g.100 g}^{-1}$) was higher than that noted with poultry manure at 28 days ($\approx$27 $\text{g.100 g}^{-1}$) by [75]. This could be explained by the very low C: N ratio of our poultry litter (C: N = 4) compared to that of poultry manure used in several reported studies (C: N ranging from 7 to 14) [72,75].

The low $C\text{-}CO_2$ quantities in S-BM microcosms compared to those in S-OWP microcosms could be explained by the high C: N ratio of BMs (ranging from 26 to 35) compared to OWPs (ranging from 4 to 16). Otherwise, DOCs in S-BM microcosms were low compared to those in S-OWP microcosms. This was due to the presence of raw forest litter (dead leaves and branches) in the BMs which contains complex organic compounds such as cellulose and lignin [76,77]. BMs would thus have a higher organic matter stability index than OWPs [78]. The influence of the biochemical composition of organic matter on carbon mineralization was demonstrated by Santos et al. [24]. These authors observed low carbon mineralization in soil + OWP mixtures containing high quantities of cellulose and hemicellulose and low concentrations of soluble compounds.

### 4.2. Impact of the Interaction between OWP and BM on Microcosm Carbon Mineralization

The positive interaction observed between OWP and BM on carbon mineralization in the microcosm could be explained by the combination of the organic carbon pools' input by OWP and BM in the microcosm in the presence of a C: N ratio conducive to carbon mineralization, and also by a synergistic effect of the OWP and BM microbial communities on carbon mineralization [79]. Indeed, the high quantities of TON (Table 1) introduced into the microcosms by the OWP would facilitate both the mineralization of OWP and BM carbon. Otherwise, each microbial community of OWP and BM involved in carbon mineralization would thus help the other communities grow and flourish [80].

### 4.3. Impact of BM on OWP Carbon Mineralization

The BM input induced the excess mineralization of OWP carbon (compared to OWP alone). This means that the BMs contained microorganisms that could access forms of organic carbon from OWPs that would be inaccessible to their normally hosted microorganisms. These included bacteria and fungi belonging to the Firmicutes, Actinobacteria, Proteobacteria, and Ascomycetes groups (Table 3), whose role in carbon mineralization has been reported [48,49,55]. Indeed, Ling et al. [52] specified that Firmicutes such as *Bacillus* and *Clostridium* (Table 3) develop early in incubation by assimilating labile carbon sources. Zhu et al. [55] observed a positive correlation between the abundance of Ascomycetes in the soil and $CO_2$ emissions. Moreover, Ling et al. and Zhang et al. [52,81] noted that Proteobacteria and Actinobacteria—via a range of enzymes (including cellulase, hemicellulase, and ligninase)—can degrade recalcitrant organic compounds.

The effect of BMs on OWP carbon varied according to the type of BM. This was due to a difference in their microbial diversity and abundance (Table 3). While all BMs induced excess C mineralization in PL, the extent was higher with LGM, followed by BG and BR. PL also had a higher DOC at incubation onset and a low C: N ratio (4), which may have benefited the mixed LGM microbial community, which was dominated by Firmicutes and Proteobacteria. The highest SS excess carbon mineralization was measured for BR—a mixture dominated by Proteobacteria and Bacteroidota. For CD, the highest excess carbon mineralization was noted in combinations with BG (or LGM)—a mixture mainly dominated by Firmicutes and Proteobacteria. Temporal variation in the effect of BM on OWP carbon would be due to a change in the microbial diversity and abundance of BM. Indeed, changes in soilborne organic matter impact the composition of bacterial and fungal communities in the short and long term [82–84]. The higher excess carbon mineralization of OWP by BMs observed between 14 and 28 days would have been due to the presence, in the same incubation period, of a high diversity and abundance of microbial communities involved in carbon mineralization.

Overall, we conclude that each of the studied BMs had a positive impact on OWP carbon mineralization. However, the highest carbon mineralization was obtained with LGM for PL; BG or LGM for CD; and BR for SS.

### 4.4. Impact of OWP or BM on Microcosm Nitrogen Mineralization

The observed change in microcosm nitrogen mineralization kinetics according to the type of OWP input (Figure 4A,C) was in line with the results of other previously reported studies [85–88]. For instance, Pansu and Thuriès [20] documented an increase in nitrogen mineralization by livestock waste input in a sandy soil incubated for 180 days, compared with soil alone. Meanwhile, Khalil et al. [32] observed nitrogen immobilization in calcareous soils mixed with wheat residues during a 90-day incubation. The Nmin quantity measured for SS was higher than that measured for PL, while PL injected more total organic N (TON) into the microcosm than SS (Table 1). PL had a lower C: N ratio than SS. Moreover, the S-PL microcosm's pH was higher than that of the S-SS microcosm during incubation (Supplementary Figure S3). Based on these different parameters (TON, C: N, and pH), there should be a higher Nmin quantity in the S-PL microcosm because a high TON quantity combined with a low C: N ratio and a high pH would jointly boost the net N mineralization [32,33,89,90]. Hence, some of the Nmin in the S-PL microcosm might have been used by the microbial communities or lost through volatilization between 7 and 28 days [91–93]. Poultry manure input in the soil is known to improve the development conditions for microorganisms [94–96] which use Nmin to meet their metabolic needs [97]. The nitrogen immobilization observed with CD throughout the incubation period (Figure 4C) could be explained by its nature (fresh), which differed from that of PL and SS (dry). Indeed, the moisture content of some OWPs at the time of their incorporation into the soil may affect nitrogen mineralization in the short term [98,99]. In addition, Santiago and Geisseler [100] reported that the low net nitrogen mineralization observed with the incorporation of fresh

broccoli residues in soil compared to dry residues was probably due to the presence of anaerobic microsites generated by the high microbial activity.

Nmin quantities of OWPs measured at 28 days also differed from those presented in other studies with the same types of OWPs and at the same date, as was also noted for carbon. For instance, the net microcosm Nmin quantity found in our sewage sludge at 28 days (10 g.100 $g^{-1}$) was lower than that reported by Levavasseur et al. [75] for sewage sludge at 28 days ($\approx$15 g.100 $g^{-1}$). This confirmed the impact of the treatment process (anaerobic digestion and total open bed drying) on the nature of our sewage sludge.

The effect of BMs on microcosm nitrogen mineralization varied with the type of BM (Figure 4B,D). LGM, which resulted in higher microcosm N mineralization, had a higher TON quantity than BG and BR and a lower C: N ratio than BG and BR (Table 1). Moreover, LGM differed from BG and BR because of its high abundance of Firmicutes, Actinobacteria, and Ascomycota, all of which are involved in nitrogen mineralization [48,49,52].

*4.5. Impact of the Interaction between OWP and BM on Microcosm Nitrogen Mineralization*

The positive/negative interactions noted between OWP and BM on microcosm N mineralization could be explained by the improvement or alteration of the conditions related to the activity of microorganisms involved in N mineralization. Indeed, the combination of organic N pools and microbial communities, and the OWP and BM input in the microcosm probably led to (i) a variation in the microcosm C: N ratio linked to the mixture, which could lead to a decrease or increase in the net mineralization compared to the theoretical mineralization; (ii) net mineralization when the OWP had a high initial DOC and low C: N ratio, thereby simultaneously enabling microbial growth and mineral N availability; and (iii) net immobilization when the high C: N ratio of the OWP was the result of microorganism competition for the N resource.

*4.6. Impact of BM on OWP Nitrogen Mineralization*

The PL-LGM, PL-BG, and PL-BR combinations increased microcosm nitrogen mineralization compared to PL between 7 and 28 days (Figure 6A). This could be explained by the increased availability of organic nitrogen in the microcosms due to the significant quantities of soluble carbon compounds introduced into the medium by PL and by a synergy between PL and BM microorganisms for nitrogen mineralization. The CD-BR combination increased the microcosm mineral nitrogen quantity (compared to CD), while CD-LGM reduced it (Figure 6B). The results are as follows: (i) CD-BR promoted N availability and stimulated microbial activity in the microcosms, while (ii) CD-LGM reduced N availability and inhibited microbial activity in the microcosms. The reduced microcosm Nmin could have been due to its immobilization by the microorganisms for their renewal or growth [101–103], or to its volatilization [20,104]. The SS-LGM and SS-BG combinations increased or decreased the microcosm mineral nitrogen quantity (compared to SS) depending on the incubation time (Figure 6C). This was probably due to a temporal variation in the diversity and abundance of microorganisms responsible for nitrogen mineralization [53,105], or to the release of nitrogen that had been immobilized following the death of microorganisms [105].

Overall, we conclude that the impact of OWP-BM combinations on microcosm nitrogen mineralization was specific to each of the combinations. The strongest synergies observed between BM and OWP for enhanced nitrogen mineralization were, thus, LGM for PL, BR for CD, and BG for SS.

## 5. Conclusions

The impacts of three BMs (LGM, BG, and BR) on carbon and nitrogen mineralization in a Senegalese Lixisol mixed with organic waste products (PL, CD, and SS) were assessed in a laboratory microcosm incubation system.

Our results demonstrated that local beneficial microorganisms (alone or combined with OWPs) impacted the microcosm carbon and nitrogen mineralization patterns. Indeed, our analysis of the microbial composition of BMs revealed the presence of several microbial

groups (Firmicutes, Actinobacteria, Proteobacteria, and Ascomycota) that are capable of mineralizing soil carbon and nitrogen. The BM input induced excess carbon mineralization in OWP compared to OWP alone; however, the highest carbon mineralization rate was obtained with LGM for PL; BG or LGM for CD; and BR for SS. We observed an increase or decrease in net Nmin quantities in the S-BM microcosms compared to the control microcosms. A comparison of the measured and calculated net Nmin quantities revealed a positive or negative interaction between OWP and BM (depending on the type of OWP, the type of BM, and the incubation time) on N mineralization. Otherwise, the PL-LGM and SS-BG combinations had the best N mineralization rates at 28 days.

Further studies should include an assessment of changes in the diversity and abundance of microbial communities in microcosms over the incubation period. This would enable an investigation of the correlation between the quantities of $C-CO_2$ and Nmin and the microcosm microbial composition over the incubation period.

The PL-LGM and SS-BG combinations were selected for field experiments on a Senegalese Lixisol, at doses of 2 t ha$^{-1}$ for PL, 6 t ha$^{-1}$ for SS, and 15 t ha$^{-1}$ for LGM or BG, to assess their agronomic and nutritional impacts on local food crops, including cowpea and orange-fleshed sweet potato.

**Supplementary Materials:** The following supporting information can be downloaded at https://www.mdpi.com/article/10.3390/agronomy13112791/s1, Figure S1: Variations in cumulated $C-CO_2$ quantities of OWP expressed in gC-CO$_2$ 100 g$^{-1}$ of organic carbon of OWP (Hypothesis 2); Figure S2: Variations in DOC quantities in microcosms expressed in mg DOC; Figure S3: Variations in microcosm pH; Tables S1a–S6c: Individual numeric values of the means, standard deviations, and *p*-value of Figure 1a to Figure 6c; Image S1: Microcosms prepared in triplicate; Image S2: $C-CO_2$ measurement by gas chromatography; Image S3: Measurement of $N-NO_3^-$ and $N-NH_4^+$ quantities.

**Author Contributions:** Conceptualization, E.N.-F., S.L., P.F. and J.-M.M.; funding acquisition, J.-M.M.; methodology, E.N.-F., S.L., P.F. and J.-M.M.; supervision, S.L., P.F., A.K., F.F. and J.-M.M.; writing—original draft, E.N.-F.; writing—review and editing, E.N.-F., S.L., P.F., L.T., K.A., A.K., F.F. and J.-M.M. All authors have read and agreed to the published version of the manuscript.

**Funding:** This research was funded by African Union Commission and European Union Commission, grant number AURG-II-2-110-2018.

**Data Availability Statement:** The authors declare that the data supporting the findings of this study are available within the main text of the manuscript and Supplementary Materials. Raw data are available from the corresponding author upon reasonable request.

**Acknowledgments:** The authors gratefully acknowledge LAMA in Dakar (UAR IMAGO, IRD, Senegal), for performing the physicochemical soil, OWP, and BM analyses prior to the incubation experiment and for measuring the mineral nitrogen forms in the incubated soil sample extracts. $C-CO_2$ measurements and the extraction of mineral N forms were carried out at the IESOL joint research laboratory in Dakar, Senegal. High-throughput sequencing was performed at ADNID (Montpellier).

**Conflicts of Interest:** The authors declare no conflict of interest.

## Abbreviations

| Abbreviation | Definition |
|---|---|
| OWP | organic waste products |
| BM | local beneficial microorganisms |
| PL | poultry litter |
| CD | cow dung |
| SS | sewage sludge |
| LGM | BM made using a forest litter collected in the Saint-Louis region and a carbon source of millet husks and groundnut shells |
| BG | BM made using a forest litter collected in the south of the Groundnut Basin and a carbon source of groundnut shells |
| BR | BM made using a forest litter collected in the south of the Groundnut Basin and a carbon source of rice bran |
| $N-NO_3$ | nitrate–nitrogen |
| $N-NH_4$ | ammonium–nitrogen |

| Nmin | net mineral nitrogen |
| C-CO$_2$ | carbon of carbon dioxide |
| OC | organic carbon |
| ON | organic nitrogen |
| TOC | total organic carbon |
| TON | total organic nitrogen |
| C: N | carbon-to-nitrogen ratio |
| DOC | dissolved organic carbon |
| OTU | operational taxonomic units |
| Assim. P | assimilable phosphorus |
| WC | water content |
| CEC | cation exchange capacity |
| DM | dry matter |

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
