# Peer review of "Local Beneficial Microorganisms Impact Carbon and Nitrogen Mineralization in a Lixisol Incubated with Organic Waste Products"

_agronomy, doi:10.3390/agronomy13112791_

Round 1

Reviewer 1 Report

Comments and Suggestions for Authors

The manuscript entitled “Local beneficial microorganisms impact carbon and nitrogen mineralization in a Lixisol incubated with organic waste products” submitted to agronomy is interesting. The parts of this manuscript are up to date and written well; however, some errors in grammar, phrases, and sentence structures were noticed throughout the manuscript. The information mentioned in the manuscript is sufficient. my only concern is the conclusion section, it is very long and can be presented shortly.

The second thing is the references sections: they need to be updated with recent citations in the discussion section.

My vote is for acceptance after minor revision.

Reviewer 2 Report

Comments and Suggestions for Authors

Given the growing awareness of the environmental repercussions of intensive agriculture, there has been a significant shift towards exploring more sustainable alternatives. Among the most promising alternatives are the use of organic waste (OWP) and the integration of biofertilizers containing indigenous beneficial microorganisms (BM) into cultivated soils. This study was designed to assess the impact of BM on carbon and nitrogen mineralization of OWP.

This study contributes to ongoing efforts to promote sustainable and environmentally conscious agricultural practices by providing a data-driven approach to improve soil health and nutrient management.

To further advance our understanding of these dynamics, future studies should consider assessing changes in the diversity and abundance of microcosm microbial communities throughout the incubation period. This would allow a more comprehensive investigation of the relationships between C-CO2 and Nmin amounts and microcosm microbial composition over time.

PL-LGM and SS-BG combinations were selected for field experiments on a Senegalese Lixisol with specific application rates aimed at evaluating their agronomic and nutritional impact on local food crops, including cowpea and orange-fleshed sweet potato. This field experimentation will provide valuable insights into the practical implications of these findings for sustainable agriculture and food production in the region.

Reviewer 3 Report

Comments and Suggestions for Authors

Noumsi-Foamouhoue et al. describes a very interesting study presents a novel insight for the most appropriate choice of organic waste products (OWP) and beneficial microorganisms (BM) mixtures for improved fertilization in sustainable production systems. Within the context of the work, it can be mentioned that the topic is relevant, and the subject is very interesting. The authors did a lot of work, and the methodology used is adequate for the objectives of the study. The results are of interest and support the conclusions. That being said, the manuscript has the potential to be accepted. However, there is still some minor issues need to be addressed before the paper could be accepted as follows:

Minor comments

Lines 35-36: keywords are already stated in the title. Please consider changing the keywords list and use synonyms.

Line 37: I suggest that the authors provide a list of abbreviations after the keywords as the manuscript is full of abbreviations.

Line 58: Please follow the journal style notation while citing references for example Santos et al. [21].

Line 67-68: Not clear

In the introduction section, the authors should shed the light on the factors affecting carbon and nitrogen mineralization in soil e.g. C/N ratio of OWP. This is a critical issue regarding the behavior of microorganisms in the soil.

Line 416: Italicize P throughout the manuscript

Kind Regards.

Comments on the Quality of English Language

Minor editing of English language required
